# Compositional Diffusion-Based
# Continuous Constraint Solvers

**Zhutian Yang**[1]   **Jiayuan Mao**[1]   **Yilun Du**[1]   **Jiajun Wu**[2]
**Joshua B. Tenenbaum**[1]   **Tomás Lozano-Pérez**[1]   **Leslie Pack Kaelbling**[1]
[1]Massachusetts Institute of Technology, [2]Stanford University

**Abstract:** This paper introduces an approach for learning to solve continuous constraint satisfaction problems (CCSP) in robotic reasoning and planning. Previous methods primarily rely on hand-engineering or learning generators for specific constraint types and then rejecting the value assignments when other constraints are violated. By contrast, our model, the compositional *diffusion continuous constraint solver* (*Diffusion-CCSP*) derives global solutions to CCSPs by representing them as factor graphs and combining the energies of diffusion models trained to sample for individual constraint types. Diffusion-CCSP exhibits strong generalization to novel combinations of known constraints, and it can be integrated into a task and motion planner to devise long-horizon plans that include actions with both discrete and continuous parameters. Project site: https://diffusion-ccsp.github.io/

**Keywords:** Diffusion Models, Constraint Satisfaction Problems, Task and Motion Planning

## 1   Introduction

Robotic manipulation planning relies critically on the ability to select continuous values, such as grasps and object placements, that satisfy complex geometric and physical constraints, such as stability and lack of collision. Existing approaches have used *separate* samplers, obtained through learning or optimization, for each constraint type. For instance, GraspNet [1] focuses on generating valid grasps, and StructFormer [2] specializes in generating semantically meaningful object placements. However, complex problems require a general-purpose solver to produce values that conform to multiple constraints simultaneously. Consider the task of 3D object packing with a robot arm, as depicted in Figure 1. Given the geometric and physical properties of the objects, our objective is to generate the target pose and motion trajectories that fulfill three essential criteria. First, all objects must be contained within the container and satisfy qualitative user requirements (e.g., bowls should be placed next to each other). Second, collisions among objects and the robot, during their movement and when stationary, must be avoided. Finally, the final configuration should be physically stable.

Constructing or training a monolithic model capable of solving *every possible combination* of goals and constraints can be challenging due to limited data. Therefore, it is essential for general-purpose robot planners to *reuse* and to *compose* individual solvers for overall tasks. A common strategy is to combine specialized methods for solving individual problems in a *sequential* manner via *rejection sampling*, which is widely used by state-of-the-art planners for object rearrangement and general task and motion planning (TAMP) problems [3, 4, 5, 6, 7]. The set of solvers is applied in a predetermined order guided by heuristics (e.g., first sampling object poses and then grasps and trajectories). If a previously sampled value violates a later constraint (e.g., the sampled pose in the box does not leave enough space for the gripper during object placement), backtracking is performed. This rejection-sampling-based approach can be highly inefficient when faced with many constraints. An alternative approach is to formulate the entire constraint satisfaction problem using a differentiable objective function and solve it using local optimization methods [8, 9]. However, these methods usually require manually-specified differentiable constraint formulas, but many important constraints, such as those corresponding to human directives like 'next to', need to be learned from data.

7th Conference on Robot Learning (CoRL 2023), Atlanta, USA.

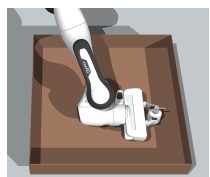 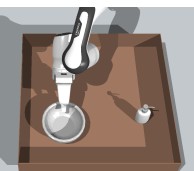 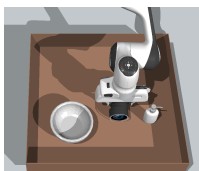 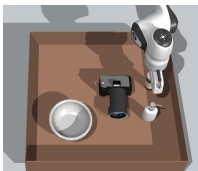 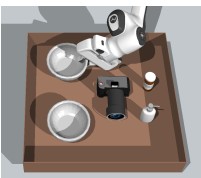

(a) The dispenser is constrained to be grasped only from the sides.

(b) The gripper pose when grasping the bowl should be reachable.

(c) The gripper at grasp poses shouldn't collide with other objects.

(d) All arm trajectories shouldn't collide with the object or the tray.

(e) The two bowls are constrained to be next to each other.

Figure 1: **Solving Continuous Constraint Satisfaction Problems by Composing Diffusion Models.** Our approach combines diffusion models, representing individual constraints, to generate object placement poses. The choice of an object's placement depends on both qualitative constraints about object placement and collision avoidance constraints on the object and the robot gripper.

We propose to use constraint graphs as a unified framework for specifying constraint-satisfaction problems as novel combinations of learned constraint types and apply diffusion-model-based constraint solvers to find solutions that jointly satisfy the constraints. A constraint graph consists of nodes representing decision variables (e.g., grasping poses, placement poses, and robot trajectories) and nodes representing constraints among these decision variables. Our method, the compositional diffusion constraint solver (Diffusion-CCSP), learns a collection of diffusion models for individual types of constraints and recombines them to address novel problem instances, finding satisfying assignments via a diffusion process that generates diverse samples from the feasible region. In particular, each diffusion model learns to generate a distribution of feasible solutions for one particular type of constraint (e.g., collision-free placements). Since the diffusion models are generative models of the set of solutions, at inference time we can **condition** on an arbitrary subset of the variables and solve for the rest. Furthermore, the diffusion process is **compositional**: since each diffusion model is trained to minimize an implicit energy function, the task of global constraint satisfaction can be cast as minimizing the global energy of solutions (in this case, simply the summation of individual energy functions). These two features introduce notable flexibility in both training and inference. Component diffusion models can be trained independently or simultaneously based on paired compositional problems and solutions. At test time, Diffusion-CCSP generalizes to novel combinations of known constraint types and graphs with more variables than those seen during training.

We evaluate Diffusion-CCSP on four challenging domains: 2D triangle dense-packing, 2D shape arrangement with qualitative constraints, 3D shape stacking with stability constraints, and 3D object packing with robots. Compared with baselines, our method exhibits efficient inference time and strong generalization to novel combinations of constraints and problems with more constraints.

## 2 Related Work

**Geometric rearrangement.** A large body of work has studied geometric rearrangement of objects in robotics subject to conditions specified in natural language [10, 11, 2, 12, 13, 14] or via formal specification or demonstration [15, 16, 17, 18, 19, 20]. Similar to our work, StructDiffusion [14] also uses diffusion models to solve for geometric rearrangements of objects. It employs a transformer-based diffusion model architecture to generate object placement poses that satisfy language-conditioned scene-level constraints, such as positioning all objects to form a circle. In comparison, our work provides a framework for solving arbitrary CCSPs that involve different constraints among objects.

**Compositional generative modeling.** Our work relates to existing work on compositional generative modeling, where individual generative models are combined to generate joint outputs [21, 22, 23, 24, 25, 26, 27, 12], primarily in the setting of image generation. Most similar to our work, Gkanatsios et al. [12] compose EBMs to represent different subgoals in compositions of object scenes. They currently only address object placements, whereas our method is generic and can combine constraints involving object poses, grasps, and robot configurations in the optimization-based sampling of object poses; this is generally more efficient than backtracking over values generated by a pose-only sampler.

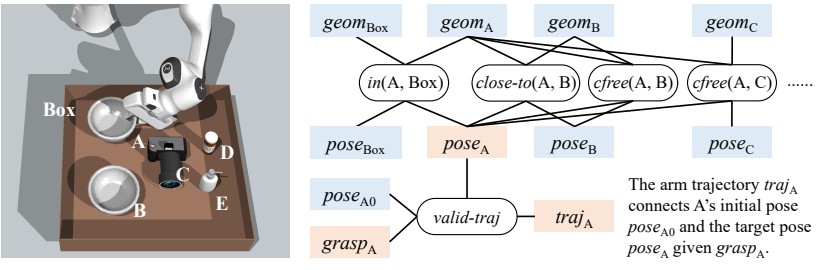

(a) Visualization of the environment while placing object A.

(b) Visualization of the constraint graphs associated with the object placement. There are three decision variables.

Figure 2: **Continuous Constraint Satisfaction Problem (CCSP) in Robot Planning.** It unifies geometric, physical, and qualitative constraints. To place A into the tray, we need to generate the grasping pose $grasp_A$, placement pose $pose_A$, and the robot arm trajectory. We omit collision-free constraints with robots in (b) for brevity.

**Task and motion planning (TAMP).** There are special-purpose 3D object packing algorithms based on heuristic-search or simulation [28, 29, 30] for finding object poses that satisfy collision and stability constraints. They are again not directly comparable to Diffusion-CCSP, because they cannot address the same range and combinations of constraints.

## 3 Compositional Diffusion Constraint Solvers

Our goal is to develop a general algorithm capable of learning and solving diverse *continuous constraint satisfaction problems* (CCSPs) encountered in robotic manipulation tasks. To achieve this, we adopt a generic graph-based representation of CCSPs unifying various geometric, physical, and qualitative constraints[*].

### 3.1 Formulation and Representation of Constraint Satisfaction Problems

Formally, a *continuous constraint satisfaction problem* (CCSP) can be represented as a graph $\mathcal{G} = \langle \mathcal{V}, \mathcal{U}, \mathcal{C} \rangle$. Each $v \in \mathcal{V}$ is a decision variable (such as the pose of an object), while each $u \in \mathcal{U}$ is a conditioning variable (such as the geometry of an object, which will be constant at performance time). Each $c \in \mathcal{C}$ is a constraint, formally a tuple of $\langle t_c, n_c \rangle$, where $t_c$ is the type of the constraint (e.g., collision-free), and $n_c = (\mathcal{V}^c, \mathcal{U}^c)$ contains two sets of variables in $\mathcal{V}$ and $\mathcal{U}$, which correspond to the arguments to this constraint. For example, a collision-free constraint between object A and B can be represented as $\langle \text{cfree}, (pose_A, pose_B, geom_A, geom_B) \rangle$. Here, $pose_A$ and $pose_B$ are decision variables that represent the target pose of objects A and B, while $geom_A$ and $geom_B$ are conditioning variables that represent the geometry of objects A and B.

Let $V$ and $U$ be assignments of values to the decision and conditioning variables $\mathcal{V}$ and $\mathcal{U}$, respectively, and let $V^c$ and $U^c$ be the subsets of assigned values to variables in constraint $c$. For example, in the 3D packing task shown in Figure 2a, the assignment $V$ to variables $\mathcal{V}$ is a mapping from the variables $pose_i$ to an $SE(3)$ pose of object $i$. The solution should satisfy that for all $c = \langle t_c, (\mathcal{V}^c, \mathcal{U}^c) \rangle \in C$, $test(t_c, (V^c, U^c)) = 1$, where *test* takes the constraint type and the assignments to each variable and returns 1 if the constraint $c$ is satisfied and 0 otherwise. The function *test* can be implemented as human-specified rules, learned classifiers, or the outcome of a physics simulator.

**Example.** Figure 2 illustrates the constraint graph associated with the pick-and-place of a bowl (A) into a box close to another bowl (B), while avoiding it and other obstacles. It includes three decision variables: $grasp_A$ (transform between object and hand), $pose_A$ (target pose), and $traj_A$ (robot trajectory). The scenario involves three constraint groups: 1) qualitative constraints, including $in(A, Box)$ ensuring the bowl's placement within the box and $close\text{-}to(A, B)$ asserting proximity between two bowls; 2) collision avoidance constraints such as $cfree(A, B)$, which asserts that objects

---

[*]Throughout the paper, we use the word qualitative constraint to refer to constraints that can not be easily and accurately described analytically—in particular, spatial relationships such as "left" and "close-to." These constraints depend on contexts and would ideally be learned from human annotations.

| **Algorithm 1** Training | **Algorithm 2** Sampling |
|---|---|
| 1: **repeat** | 1: $V_T \sim \mathcal{N}(\mathbf{0}, \mathbf{I})$ |
| 2:  $\mathcal{G}, V_0 \sim \mathcal{D}$ | 2: **for** $t = T, \ldots, 1$ **do** |
| 3:  $t \sim \text{Uniform}(\{1, \ldots, T\}); \boldsymbol{\epsilon} \sim \mathcal{N}(\mathbf{0}, \mathbf{I})$ | 3:   **if** $t > 1$ **then** $\mathbf{z} \sim \mathcal{N}(\mathbf{0}, \mathbf{I})$ **else** $\mathbf{z} = \mathbf{0}$ |
| 4:  Take gradient descent step on | 4:   $V_{t-1} = \frac{1}{\sqrt{\alpha_t}} \left( V_t - \frac{1-\alpha_t}{\sqrt{1-\bar{\alpha}_t}} \sum_{c \in \mathcal{C}} \epsilon_{t_c} \left( V_t^c, U^c, t \right) \right) + \sigma_t \mathbf{z}$ |
| 5:    $\nabla_\theta \left\| \epsilon - \sum_{c \in \mathcal{C}} \epsilon_{t_c} (\sqrt{\bar{\alpha}_t} V_0^c + \sqrt{1-\bar{\alpha}_t} \epsilon, t)) \right\|^2$ | 5: **return** $V_0$ |
| 6: **until** converged | |

A and B do not collide in the final placement; 3) trajectory constraints, encapsulated by *valid-traj*, which assert that the trajectory $traj_A$ is a feasible robot path that connects the bowl's initial position $pose_{A0}$ and the target $pose_A$, given $grasp_A$. It is important to note that these constraints are correlated: for example, the choice of $pose_A$ requires the consideration of the robot trajectory to ensure the existence of a valid grasping pose and trajectory.

## 3.2 Compositional Diffusion Models

Given a set of constraints $\mathcal{C}$, decision variables $\mathcal{V}$, and conditioning values $\mathcal{U} = U$, we wish to find an assignment of $V_0$ that satisfies $\mathcal{C}$. We represent the conditional distribution of variable assignments given an individual constraint $c$ using a diffusion model $p_c(\mathcal{V}^c = V^c \mid \mathcal{U}^c = U^c) \propto \mathbb{1}[test(t_c, (V^c, U^c))]$, which is effectively uniform over the (bounded) region of $\mathcal{V}^c$ that satisfies the constraint. Finding a satisfactory assignment of variables $V_0$ then corresponds to finding an assignment that maximizes the likelihood in the joint distribution constructed from the product of the individual constraint models. Each constraint diffusion model represents $p(\mathcal{V}^c \mid \mathcal{U}^c)$ in the form of an energy-based model, $p(\mathcal{V}^c \mid \mathcal{U}^c) \propto e^{-E(\mathcal{V}^c \mid \mathcal{U}^c)}$ through its score [24, 26], so the probability maximization problem corresponds to the energy minimization problem:

$$V_0 = \arg\max_V \prod_{c \in \mathcal{C}} p_c(\mathcal{V}^c \mid \mathcal{U}^c) = \arg\min_V \sum_{c \in \mathcal{C}} E(\mathcal{V}^c \mid \mathcal{U}^c) .$$

We solve this energy minimization problem by using a sampler based on the annealed unadjusted Langevin algorithm (ULA) [31, 26]. The ULA algorithm is a special case of Langevin Monte Carlo — it iteratively optimizes a noisy sample using the gradient of the energy function, with added noise at each step of optimization.

During training, we optimize a noise prediction model $\epsilon_t$ for each constraint type $t$. Each $\epsilon_t$ takes conditioning variables $U^c$ and a noisy version of $V^c$ as input, and predicts the noise applied on $V^c$. Most simply, we could train the diffusion models independently for each constraint, and combine them at prediction time. In our experiments, we train multiple diffusion models simultaneously: since each training example corresponds to some constraint graph, it is natural to train all the constraints that are instantiated in that graph jointly from the example. Note that the constraint graphs within the training set and between training and testing need not be the same—they just need to be constructed from the same basic set of constraint types.

Specifically, our training dataset $\mathcal{D}$ contains a set of constraint graphs, each with conditioning values and a solution to the decision variables: $\langle \mathcal{G} = (\mathcal{V}, \mathcal{U}, \mathcal{C}), U, V_0 \rangle$. For each type of constraint, we randomly initialize a denoising function $\epsilon_t$, and sum the denoising predictions over the constraints in the graph to denoise a noisy version of $V_0$:

$$\mathcal{L}_{\text{MSE}} = \mathop{\mathbb{E}}_{\substack{\langle \mathcal{G} = (\mathcal{V}, \mathcal{U}, \mathcal{C}), U, V_0 \rangle \in \mathcal{D} \\ \boldsymbol{\epsilon} \sim \mathcal{N}(\mathbf{0}, \mathbf{I}), t \sim \text{Uniform}(1, \cdots, T)}} \left[ \left\| \epsilon - \sum_{c \in \mathcal{C}} \epsilon_{t_c} \left( \sqrt{\bar{\alpha}_t} V_0^c + \sqrt{1-\bar{\alpha}_t} \epsilon, U^c, t \right) \right\|^2 \right],$$

where $\epsilon$ is random Gaussian noise applied to $V_0$, $\alpha_t$ are predefined noise scheduling for training and sampling following Nichol and Dhariwal [32], $\bar{\alpha}_t := \prod_{i=1}^t \alpha_i$, $t$ is a randomly sampled diffusion step, and $T$ is the total number of diffusion steps. The full pseudocode for training and sampling from our conditional diffusion models for constraints can be found in Algorithms 1 and 2.

## 3.3 Geometric and Physical Constraint Solving with Diffusion-CCSP

Given the general formulation of Diffusion-CCSP, we describe its concrete implementation in our robot planning setting. Recall that for each type of constraint, we have defined a "denoise" function $\epsilon_t$

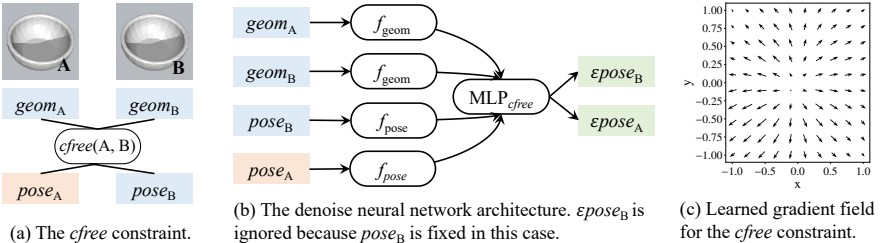

(a) The *cfree* constraint.

(b) The denoise neural network architecture. $\varepsilon pose_B$ is ignored because $pose_B$ is fixed in this case.

(c) Learned gradient field for the *cfree* constraint.

Figure 3: **Illustration of the denoising neural network** for satisfying constraint *cfree* (collision-free) and the gradient map learned by Diffusion-CCSP for the centroid of B at $(x, y)$ when A is at $(0, 0)$.

that takes the conditioning values and (the noisy version of) the decision values as inputs, and predicts the noise applied on the decision values. In our robotic domains, we focus on two variable types: geometry variables and pose/transformation variables. We illustrate the neural network architecture for collision-free constraints though the construction easily generalizes to other constraint types.

Recall that the argument to a collision-free constraint $c$ has four elements: $\mathcal{U}^c = (geom_A, geom_B)$, $\mathcal{V}^c = (pose_A, pose_B)$. We use $pose_A^t$ to denote the value of this variable at diffusion time step $t$. The output of $\epsilon_{\text{cfree}}$ is a tuple $(\epsilon pose_A^t, \epsilon pose_B^t)$ which corresponds to the noise applied on $pose_A$ and $pose_B$ at diffusion time step $t$. In our implementation, an object's shape is represented as a 3D axis-aligned bounding box, and its pose is represented as a 5-dimensional feature vector composed of the 3-DoF translation and rotation $\gamma$ about $+z$ axis, encoded using $\sin(\gamma)$ and $\cos(\gamma)$. We use a multi-layer perceptron (MLP) $f_{\text{geom}}$ to encode the object shapes (shared across all constraints) and a constraint-specific MLP $f_{\text{pose}}$ to encode poses. Both geometric features and pose encoders map input representations into a fixed-length vector embedding. The function $\epsilon_{\text{cfree}}$ is implemented as

$$\epsilon_{\text{cfree}}^t(geom_A, geom_B, pose_A, pose_B) = g_{\text{cfree}}\left(f_{\text{geom}}(geom_A) \oplus f_{\text{geom}}(geom_B) \oplus f_{\text{pose}}(pose_A) \oplus f_{\text{pose}}(pose_B) \oplus t\right),$$

where $\oplus$ denotes vector concatenation, and $g_{\text{cfree}}$ is another MLP model that takes the concatenated input-feature vectors and outputs a vector of length 10. The first 5 entries correspond to the prediction of $\epsilon pose_A^t$, and the rest correspond to $\epsilon pose_B^t$, where $t$ is the decoding time step.

**Example.** Figure 3 illustrates the neural network architecture for one specific constraint and the corresponding gradient map learned by Diffusion-CCSP. In this example, our objective is to generate a pose for object A that avoids collisions with object B in its current position. The neural network encodes the four input features and predicts the noise for $pose_A$ and $pose_B$. As the pose of the object B remains unchanged, the noise (gradient) predicted by $\epsilon_{\text{cfree}}$ for its pose is disregarded. Figure 3c visualizes the gradient map for the X and Y components of $pose_A$, where the gradient points approximately outward from the current location of object B. It is important to note that the gradient over the pose variable $pose_A$ will have contributions from multiple constraints in the constraint graph. See the gradient fields of all qualitative constraints in Appendix F.

**Planning.** So far, we have presented a generic solution to solving constraint satisfaction problems that involve geometric and physical constraints, assuming that the constraint graph is given as input to the algorithm. This approach can be directly used to solve particular tasks such as the pose prediction task in object rearrangements. Our primary motivation is to solve subproblems that arise during general robot manipulation planning. In Appendix E, we illustrate how Diffusion-CCSP can be integrated with a search algorithm to solve general task and motion planning (TAMP) problems, where the constraint graphs are automatically constructed based on the sequence of actions that has been applied and the goal specification of the task.

## 4 Experiments

We evaluate Diffusion-CCSP through CCSP tasks incorporating geometric, physical, and qualitative constraints. We include all environmental details and sample complexity discussions in Appendix A. For implementation and training details and additional discussions on model training and inference, see Appendix B. For analysis of Diffusion-CCSP's failure cases in each task, see Appendix C.

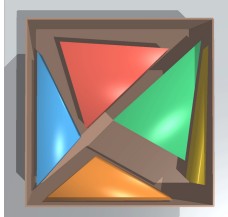

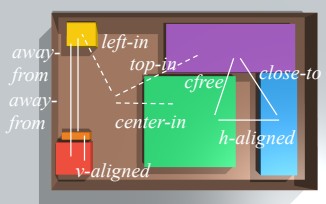

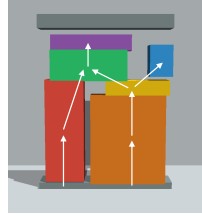

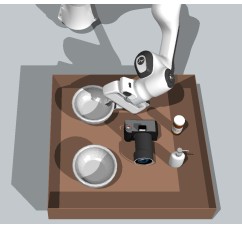

| (a) 2D Triangle Packing. | (b) 2D Shape Arrangement with Qualitative Constraints | (c) 3D Object Stacking with Stability Constraints. | (d) 3D Object Packing with Robots. |

Figure 4: **Illustration of four domains.** (a) Triangle packing. (b) Dense 2D packing with qualitative constraints. The figure shows a subset of 45 constraints of 13 types. (c) 3D object stacking. The arrows show the support relationships. (d) 3D object packing with a panda robot.

We compare our full model to one model variant, a sequential-sampling baseline, and StructDiffusion [14]. ***Diffusion-CCSP w. Reverse Sampling***: In this variant, we directly use the standard reverse diffusion process to sample from composed diffusion models [24]. This method is faster at sampling than ULA, but is generally found to have worse generalization performance. We include a detailed comparison between these two in Appendix D. ***Sequential Sampling***: We sequentially sample each decision variable according to a generic sampler (e.g., for 2D poses, we randomly sample a location that is within the tray and a random rotation) and check all geometric constraints (e.g., *in* and *cfree* associated with the decision variable. For each variable, we sample 50 samples, and report failure when no successful samples can be produced within 50 samples. ***StructDiffusion***: StructDiffusion is a transformer-based diffusion model for predicting object placement poses based on object shapes and a single scene-level constraint. It takes a set of object shape representations, the noisy sample of their poses, and the diffusion time step, and predicts the noise on object poses. Since it can only handle a single global constraint, it is only applicable in two of our tasks (Section 4.1 and Section 4.4).

## 4.1 2D Triangle Packing

The first domain considers packing a set of random triangles into a square tray, ensuring no overlap. Triangles can be rotated, and their shapes are encoded by their vertex coordinates in their zero poses. The model's output includes a 2D translation vector and a 1D rotation. Training data consists of 30,000 solutions to problems with 2-4 triangles, and testing includes 100 problems with 2-5 triangles. We generate training solutions by randomly splitting the tray.

Fig. 5a shows the result, and Fig. 4a shows a concrete testing problem and its solution found by Diffusion-CCSP. When the number of triangles becomes large, it becomes increasingly difficult for the rejection-sampling baseline to succeed. The transformer architecture used in StructDiffusion successfully models the relational structure among all pairs of objects. Therefore, it performs similarly to our method and has a better generalization to out-of-distribution CCSPs in this task.

## 4.2 2D Shape Arrangement with Qualitative Constraints

This domain involves packing rectangles into a 2D box, satisfying geometric and qualitative constraints resembling scenarios such as dining table settings and office desk arrangements. Qualitative constraints include *center-in*(A, Box), *left-in*(A, Box), *left-of*(A, B), *close-to*(A, B), *vertically-aligned*(A, B), etc., along with the geometric ones such as containment (*in*) and collision-free (*cfree*). Different problem instances have different box sizes and constraints. The training dataset consists of 30,000 problems and solutions with 2 to 4 objects, while the testing set contains 100 problems with 2 to 5 objects. Training examples are generated by randomly splitting the tray.

Fig. 5b shows the result, and Fig. 4b shows a concrete test problem that involves six objects and 45 constraints (including qualitative constraints shown in the figure and geometric constraints such as *in* and *cfree*). The task is very challenging when the number of objects and the number of constraints becomes large. Therefore, the baseline rejection sampling method barely solves any problem instances with 6 objects. Diffusion-CCSP significantly outperforms the baselines. We also

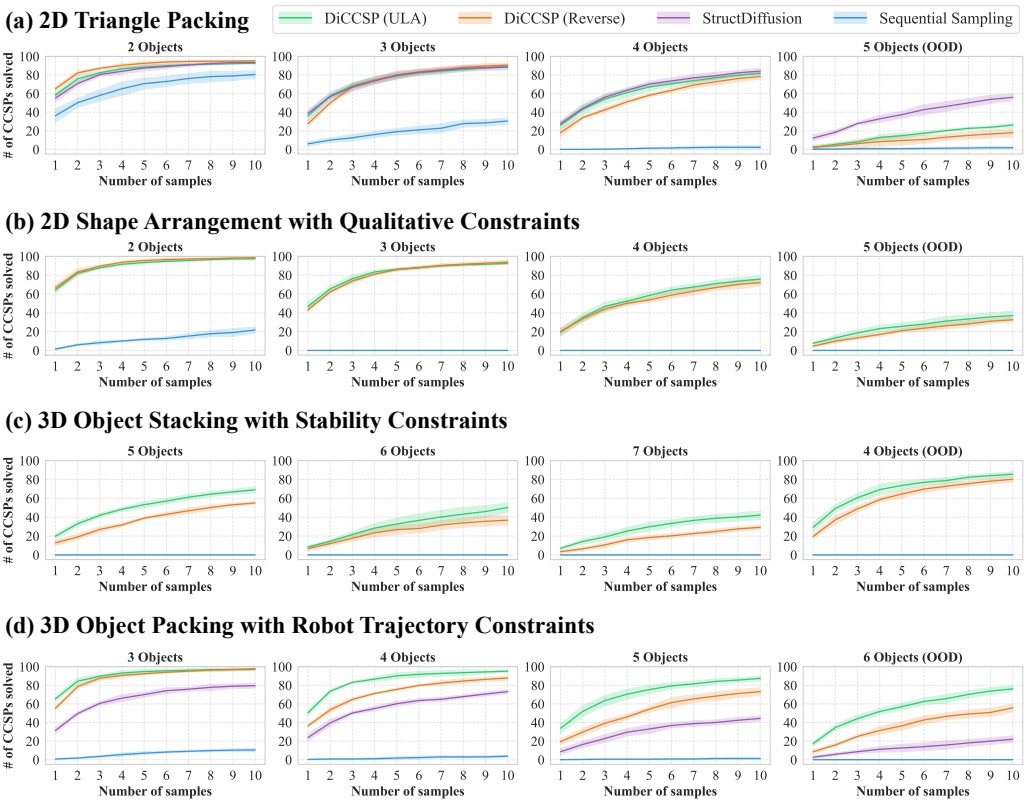

Figure 5: **Quantitative Comparisons of Constraint Solvers.** Accumulated number of problems solved in 10 runs of different models. OOD=Out of training distribution. The shaded area indicates the standard deviation of various models across five different seeds. The sequential sampling baseline completely failed for task (c) and hard tasks in (b) and (d). Our full model performs better than a variant without ULA sampling and better than StructDiffusion in more complex problems (d).

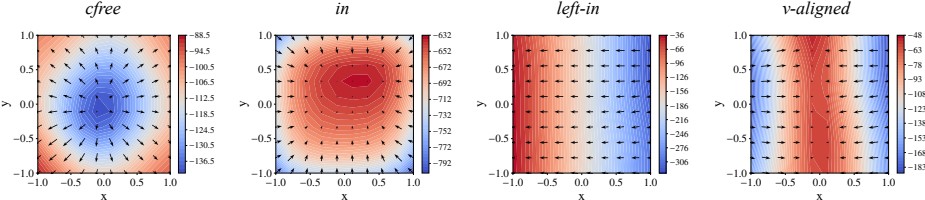

Figure 6: Visualization of learned energy functions of selected geometric and qualitative constraints.

visualized the learned energy landscape for a few constraints in Fig. 6 learned by our model, and more visualizations in Appendix F.

## 4.3  3D Object Stacking with Stability Constraints

The third domain considers arranging trapezoidal prisms onto a shelf to satisfy a given support structure and stability. The goal is to generate a sequence of object placement actions (including object names and their target poses). This reflects real-life tasks such as organizing storage containers, which often involve tilted boxes due to varying prism heights. All of our training and testing problems involve at least one "bridge-like" structure, i.e., there is at least one object supported by multiple objects simultaneously. Furthermore, since the output of models is a sequence of object placements, a sequence is successful only if the final stage and *all intermediate states* are physically stable. We randomly sample the sizes of the shelf region and the shapes. We use a physical simulator to generate training examples. The training dataset has 24,000 problem instances and solutions with 5 to 7 objects, and the testing set includes 100 problem instances with 4 to 7 objects.

| Objects | N=4 | N=5 |
|---|---|---|
| # of Calls to Diffusion-CCSP | 11.9 | 33.33 |

| Objects | N=2 | N=3 | N=4 | N=5 | N=6 |
|---|---|---|---|---|---|
| # of Calls to Diffusion-CCSP | 1.33 | 1.97 | 2.96 | 4.37 | 7.26 |

(a) 3D Object Stacking.  (b) Task 3: 3D Object Packing with Robots.

Table 1: Average number of calls to Diffusion-CCSP in order to solve the full TAMP problem.

An example of the task is illustrated in Fig. 4c. Given that neither our algorithm nor the baselines specifically plan for object placement orders, we integrate them into a very simple task and motion planner, as detailed in Appendix E. This planner randomly samples object placement orders and box "support" structures (i.e., the arrangement of objects supporting other objects), then invokes the Diffusion-CCSP solver to find solutions. We assess two metrics: first, the performance of Diffusion-CCSP with a fixed CCSP graph, illustrated in Fig. 5c; second, the number of calls to Diffusion-CCSP required to solve problems involving varying quantities of objects, as shown in Table 1a. For instance, packing 6 boxes into a cabinet necessitates, on average, 25 different object placement and support structure samples. If a CCSP is not solvable by Diffusion-CCSP, it might truly be infeasible due to incorrect sampled placement orders or support structures. Therefore, Table 1a shows the integrated system's overall performance. Notably, given that we can process multiple CCSPs as a GPU batch and that generating object orders is inexpensive, the system does not have high latency overall. In a batch of 100 CCSPs, it takes on average 0.01-0.04 seconds for Diffusion-CCSP (Reverse) to solve each CCSP, and 0.06-0.19 seconds for Diffusion-CCSP (ULA).

### 4.4 Task 4: 3D Object Packing with Robot Trajectory Constraints

This domain involves packing 3D objects into a box container in a simulated environment using PyBullet, with a Franka Emika Panda arm. The dataset includes 53 objects from the PartNet Mobility dataset and ShapeNet [33, 34], 15 of which only afford side grasps, while others can be grasped from the top. We pre-generate a set of grasps for each object, and use a motion planner to find robot trajectories. The task and motion planner and Diffusion-CCSP jointly solve for grasping choices, object poses, and arm motions. Although our model doesn't directly generate grasps or trajectories, it reasons about object geometries and has learned to predict poses for which it is likely that there are collision-free trajectories (which is later verified with a motion planner.) The training dataset has 10,000 problem instances and solutions with 3 to 5 objects, and the testing set includes 100 problem instances with 3 to 6 objects. Diffusion-CCSP successfully finds solutions to 60-80 percent of the problems in just 10 samples for the most challenging 6-triangle packing problems.

Fig. 5d illustrates the number of samples needed for tasks involving a different number of objects. Overall, Diffusion-CCSP significantly outperforms the rejection-sampling-based solver and Struct-Diffusion, especially for larger numbers of objects. Diffusion-CCSP can even generalize directly to six-object packing problems without additional training. Table 1b showcases the full pipeline's performance, which includes generating placement orders, grasps, poses, and trajectories.

## 5  Limitation and Conclusion

**Limitations.** All constraints we have explored in this paper have a fixed arity (i.e., they involve a fixed number of objects). An interesting direction is to consider variable-arity constraints. Another future improvement is to incorporate more sophisticated shape encoders and learning constraints from real-world data, such as online images, for a wider range of applications.

**Conclusion.** In conclusion, this paper proposes a learning-based approach that leverages the compositionality of continuous constraint satisfaction problems (CCSP) and uses diffusion models as compositional constraint solvers. The proposed algorithm, namely the compositional diffusion constraint solver (Diffusion-CCSP), consists of modular diffusion models for different types of constraints. By conditioning these models on subsets of variables, solutions can be generated for the entire CCSP. The model exhibits strong generalization capabilities and can handle novel combinations of known constraints, even with more objects and constraints than encountered during training. It can also be integrated with search algorithms to solve general task and motion planning problems.

**Acknowledgements.** This work is in part supported by NSF grant 2214177, AFOSR grant FA9550-22-1-0249, FA9550-23-1-0127, ONR MURI grant N00014-22-1-2740, ARO grant W911NF-23-1-0034, the MIT-IBM Watson AI Lab, the MIT Quest for Intelligence, the Boston Dynamics Artificial Intelligence Institute, the Center for Brain, Minds, and Machines (CBMM, funded by NSF STC award CCF-1231216), the Stanford Institute for Human-Centered Artificial Intelligence (HAI), and Analog Devices, JPMC, and Salesforce. Any opinions, findings, and conclusions or recommendations expressed in this material are those of the authors and do not necessarily reflect the views of our sponsors.

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

# Supplementary Material for
# Compositional Diffusion-Based Continuous Constraint Solvers

Appendix A describes how our datasets are collected for each task and how the inputs are encoded for diffusion models. We also provided discussions on sample efficiency of Diffusion-CCSP on the tasks studied. Appendix B details hyper-parameter choices of our models, implementation details of StructDiffusion baseline [14], as well as training details for both. We have also included additional discussions on model training and fine-tuning, constraint weights, local optima in optimization, and partial observability. Appendix C discusses common failure cases in solutions generated by out Diffusion-CCSP. Appendix D includes a detailed experiment on the number of samples it takes for Diffusion-CCSP to solve CCSP problems in each task. Appendix E discusses how to integrate Diffusion-CCSP for CCSP solving with task and motion planning (TAMP) algorithms. Appendix F shows the learned energy landscapes for qualitative constraints in 2D shape arrangement tasks.

## A  Problem Domains and Data Generation

In all datasets, the number the examples involving different number of objects are the same.

### A.1  2D Triangle Packing

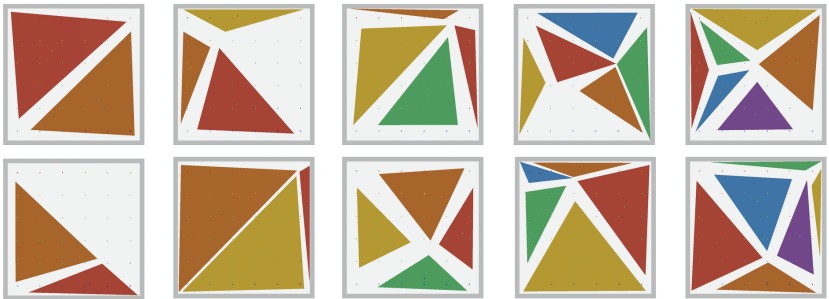

Figure 7: Example collision-free configurations of triangles

**Data.** The data is generated by randomly sampling points in the square and, connecting them with each other and the edge points using the Bowyer–Watson algorithm, then shrinking each triangle a little to make space among them.

**Encoding.** We define the resting pose of a triangle as the pose when the vertex $A$ facing the shortest side $BC$ is at origin and its longest side $AB$ is aligned along $+x$ axis. Its geometry is encoded using three numbers, the length $|AB|$ and the vector $(x, y) = \overrightarrow{AC}$. Its pose is encoded as the 2D coordinates of vertex $A$, along with the *sin* and *cos* values of the rotation $\theta$ from the testing pose.

### A.2  2D Shape Arrangement with Qualitative Constraints

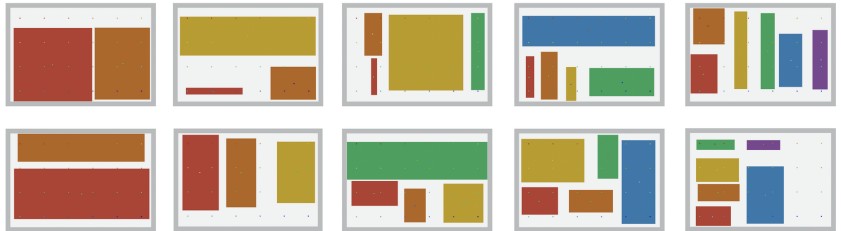

Figure 8: Example rearrangements of rectangles with 11 types of qualitative constraints.

**Data.** The data is generated by recursively splitting the tray at different proportions until depth 3. For each resulting region, a random padding is added to each side. Regions whose area or side is too small

are discarded. Labels of qualitative constraints are created by hand-crafted rules, e.g. 'close-to' means the distance between two objects is smaller than the maximum width of two objects. All qualitative constraints include 'center-in', 'left-in', 'right-in', 'top-in', 'bottom-in', 'left-of', 'top-of', 'close-to', 'away-from', 'h-aligned', and 'v-aligned'.

**Encoding.** The resting pose of a rectangle is when its longer side is oriented horizontally. Its geometry is encoded using its width and length at resting pose. Its pose is the 2D pose of the centroid, along with the $sin$ and $cos$ encoding of the object's $yaw$ rotation. Many problems require some objects to be at vertical positions in order for all the constraints to be satisfied.

### A.3    3D Object Stacking with Stability Constraints

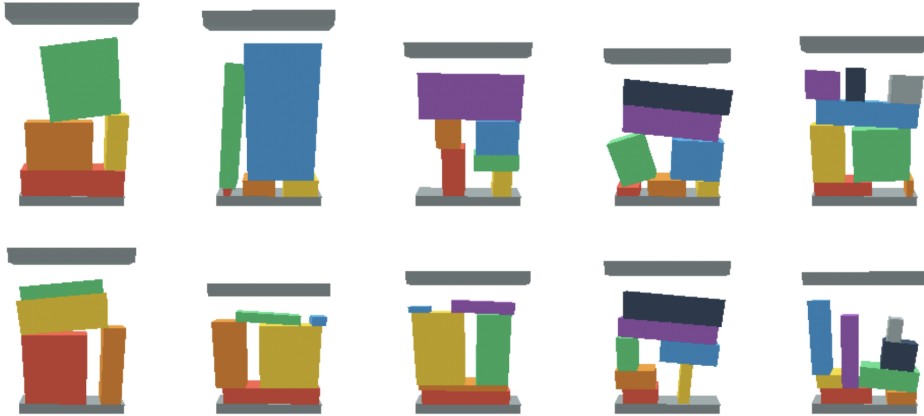

Figure 9: Example stable configurations of rectangles, with `in`, `cfree`, and `supported-by` constraints. There is at least one object that's supported by multiple objects.

**Data.** The data is generated by randomly splitting a 2D vertical region, then shrinking and cutting each region. The resulting cuboids are initiated in PyBullet simulator and letting them drop until rest. We filter for configurations where the final state is stable. Then all objects are removed one by one to test that each intermediate configuration is also stable. An upper shelf is added to be just enough to fit the current configuration with a little overhead space.

**Encoding.** The resting pose of a cuboid is when its longer side is oriented horizontally. The cuboid's geometry is encoded using its width and length. The cuboid's pose is the 2D pose of the centroid, along with the $sin$ and $cos$ encoding of the cuboid's $roll$ rotation.

### A.4    3D Robot Packing with Robots

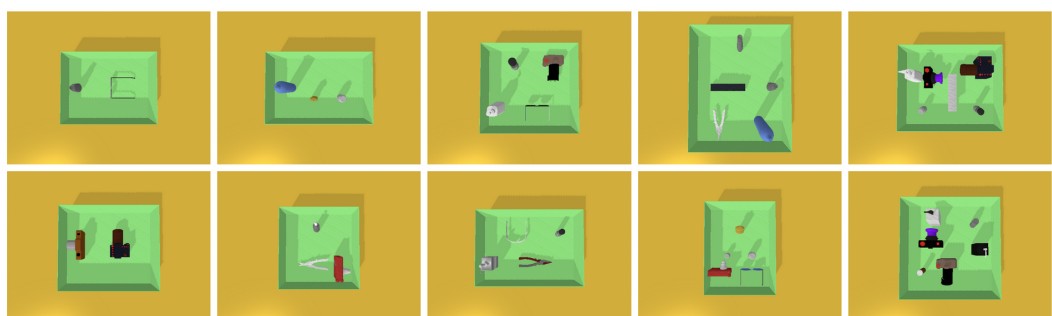

Figure 10: Example configuration of 3D shapes that enables the robot place each object in a given sequence without colliding into other objects already placed. All the bottles, dispensers, and bowls affords only side grasps.

**Data.** The 10 categories of objects used in this dataset include 'Dispenser', 'Bowl', 'StaplerFlat', 'Eyeglasses', 'Pliers', 'Scissors', 'Camera', 'Bottle', 'BottleOpened', and 'Mug', contributing in total 53 object assets. For each asset, a fixed number of grasps are generated beforehand that points directly to one of its five faces, e.g. $+x, -x, +y, -y, +z$. The problems are generated by randomly splitting the tray into rectangle regions, then fitting into each region an object with random category, asset, and scale. An order to place the objects is sampled, then a TAMP planner is called to check whether the corresponding grasps can be found (by enumerating all combinations of order and grasps) so that the gripper at each grasp pose won't collide with any objects already in the goal region and all arm trajectories going to the grasp configurations can be found.

**Encoding.** The objects' geometry is encoded using bounding box (e.g., width, length, height). An object's grasp is encoded using a five-dimensional vector, indicating the face that the gripper points to. For example, $[0, 0, 0, 0, 1]$ is a top grasp. The object's pose is encoded using its 3D coordinate in the tray frame, along with the $sin$ and $cos$ encoding of the object's $yaw$ rotation.

### A.5 Discussion: Sample Complexity of Diffusion-CCSP

Our training datasets are on the scale of 10k-30k for each task. It is always important to be aware of the amount of training data needed and the cost of obtaining it from various sources. We observed that we can train the models in simulation with abundant data and believe they can be applied to real-world robotic tasks. In our system, we learn two categories of constraints. First, geometric and physical constraints are independent both of the robot (and hence of perceptual and motor noise and irregularities that cause most sim-to-real difficulties) and of any human input. Learning to check that object placements are collision free, or that robot configurations are kinematically feasible can be done entirely via data obtained in simulation and transfer without difficulty to the real world. Second, qualitative constraints corresponding to human input must be obtained from human annotation. Such annotations are costly obtain and, as a matter of future work, we would like to leverage additional data sources, such as object relationship information in large vision/language models.

### A.6 Discussion: Partial Observability

Across all experiments in the paper, we have been assuming a fully-observed setting where we know object shapes (in particular, their bounding boxes) and their initial 3D poses. We think one possible way to handle at least some types of partial observability in our setting is through replanning. Taking object packing as a concrete example, given the segmented (partial) point clouds for objects, we use learning or heuristic methods to complete the object mesh and then run our solver to compute plans. During execution, assuming we can sense object collisions while moving them, we can always replan for a different placement in a closed-loop manner.

## B  Implementation and Training Details

We used 1000 diffusion timesteps for all models, except for Diffusion-CCSP on Task 4 robot packing, which used 500 diffusion timesteps. In neural network input encoding, all object dimensions and poses are normalized with regard to that of the container or shelf region.

***Diffusion-CCSP***: Given the geometry, pose, and grasp features, we use geometry, pose, and grasp encoders to process them to the same hidden dimension of 256, which is the same dimension as time embedding. The outputs from individual diffusion models have the same dimension as input pose embedding. The output for each object's pose are composed by taking the average of all outputs of diffusion models that have the object as an argument. They are processed through a pose decoder to get the output reconstructed noise value added to the poses. The encoders, pose decoder, and diffusion models are trained together using the L2 loss between reconstructed noise value and input noise value, as detailed in Section 3.2. For our ULA variant, we used sampling steps of 10 and step sizes of $2\beta(t)$ where $\beta$ is cosine beta schedule for the diffusion model.

***Rejection-Sampling***: The objects are sampled in sequence inside the container (collision-free with the container is guaranteed). For each object, its pose is sampled uniformly at random at most 50 times to find a placement that's collision-free to all objects already placed.

***StructDiffusion***: We implemented the architecture of concatenating the geometry and pose embeddings for each object as one token (dimension 512 for each token) to the transformer encoder, plus positional encoding and time encoding. For task 4, we also concatenated the chosen grasp embedding to the object embedding (total dimension 768 for each token). We used four layers of residual attention blocks and two heads. We added normalization layers before and after the transformer layers. We used the same diffusion timesteps as our Diffusion-CCSP models. Because there is no language input, we omitted the word embeddings from the architecture.

## B.1 Discussion: Individual Constraint Diffusion Training and Finetuning

Our diffusion models for individual constraints can be trained either jointly or independently; however they are trained, they can be applied in novel combinations at performance time. In our experiments, we concurrently train all constraints since our data naturally encompasses multiple constraints in a single scene, such as object collision-free constraints in a bin packing problem. It is also possible to finetune a pretrained constraint solver with new data. In some scenarios such as generalization to unseen object shapes this might be very helpful.

## B.2 Discussion: Weights for Energy Function Composition

In general, selecting appropriate weights for different constraints is important and challenging, which has been observed in Urain et al. [35] and Liu et al. [24]. In our experiments, we consistently use weight 1 for all constraints, and this setting has proven effective in our tasks. We postulate two reasons for this. First, visualizations of the learned energy functions (see Appendix F) indicate that the energy field is steep. The areas in the configuration space that meet the constraints exhibit considerably lower energy than areas that don't. As such, when constraints are combined, the solver tends to prioritize solutions that satisfy all constraints. Second, during training, our models are jointly trained to explain data with various combined constraints. This implicitly forces the diffusion models to learn score functions that support combinations.

## B.3 Discussion: Local Optima in Optimization

Bad local optima can be an issue for multi-constraint optimization problems. In this paper, we partially address this problem by running multiple seeds in parallel. That is, given a CCSP problem, we run the reverse diffusion process with multiple randomly sampled initial noise values. For instance, in the 3D object stacking task, on average, 33.3 samples are needed for solving tasks involving 5 objects. This does not significantly affect our runtime because different random seeds can be run in parallel. In two cases shown in the table (object packing and stable stacking), we have used the physical simulator to check solution validity—whether objects are in collision-free poses and whether the system is physically stable. In other cases where solution checkers are unavailable, we can consider learning classifiers for conditions.

## C  Failure Case Analysis

Example failure cases for Task 2 involving qualitative constraints are shown in Figure 11. One insight we gained is that *the failure rate to satisfy a qualitative constraint is inversely correlated with the area of its defining region*. We ran a Diffusion-CCSP 4000 times to solve a test set of 400 CCSPs involving 2-4 shapes and counted the number of unsatisfied constraints in solutions generated by Diffusion-CCSP. From Figure 12a, we make two observations: (1) the failure rate is inversely correlated with the amount of data for each constraint, and (2) constraints with discontinuous regions are harder to satisfy (e.g. v-aligned, close-to). Then, we look at a problem with one object A (size 0.5 by 0.5) in the center of a container. We estimate the area of the defining region of each constraint

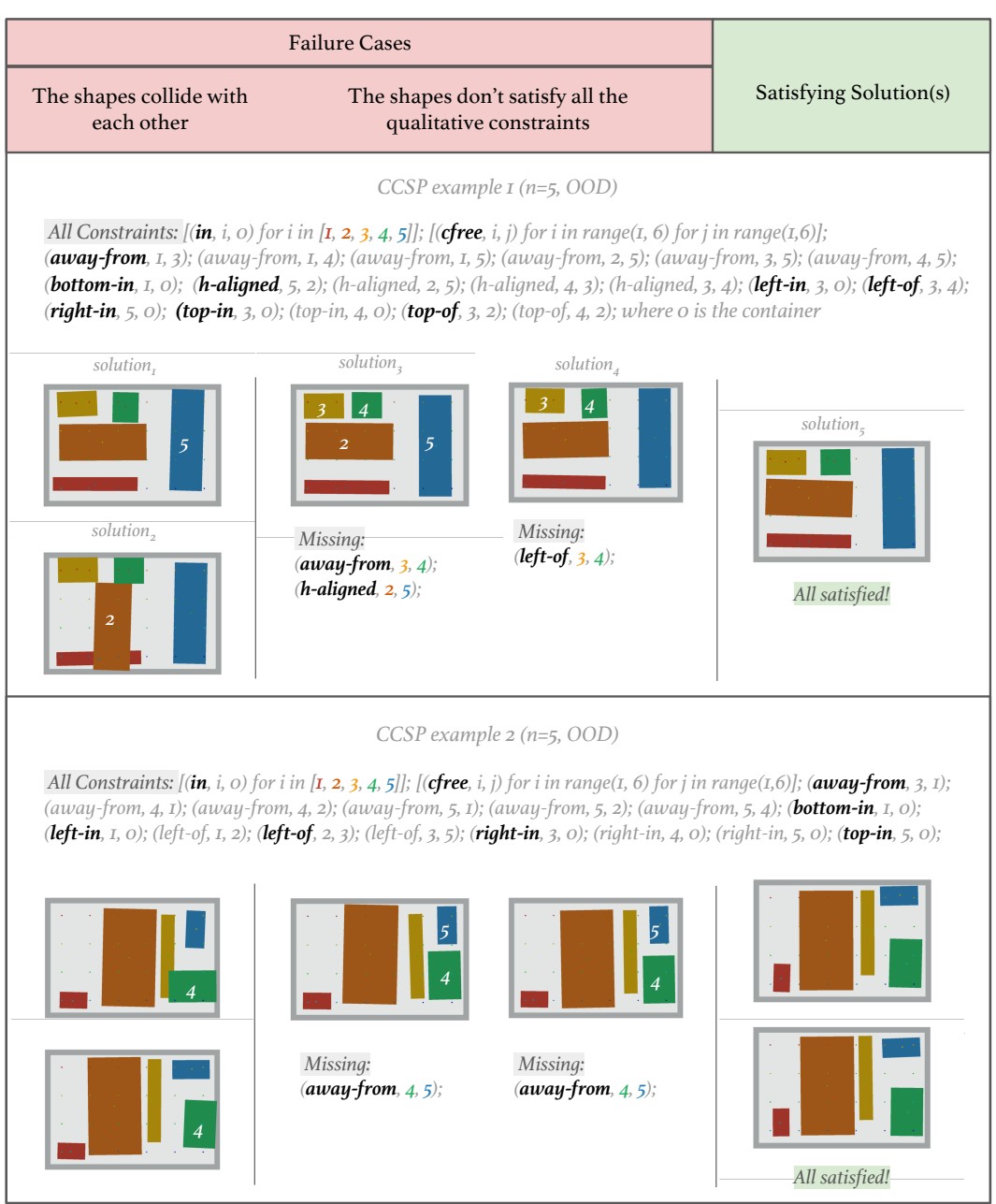

Figure 11: **Failure and success samples for Task 2: 2D Shape Rearrangement with Qualitative Constraints**. All solutions are generated by Diffusion-CCSP (ULA), trained on 30k data up to 4 objects.

by doing rejection sampling on the pose $(x, y)$ of object B (size 0.5 by 0.5) that doesn't collide with A (except for *center-in*). From Figure 12b, the order is highly similar to the order in Figure 12a, supporting the hypothesis that the smaller the defining region of a constraint, the harder it is to satisfy. Note that this experiment assumes two object. Therefore, although the area is large for *cfree* in this estimate, it doesn't represent the dataset where there are 2 - 5 shapes in each problem. So we exclude it from further discussions.

Example failure cases for Task 1 and 3 are shown in Figure 13 and Figure 14.

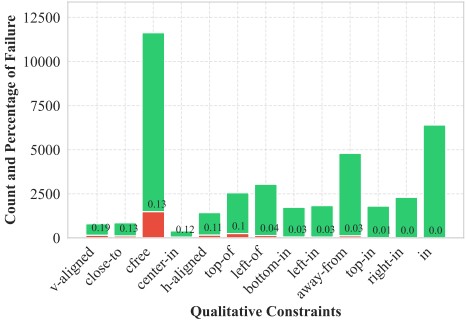 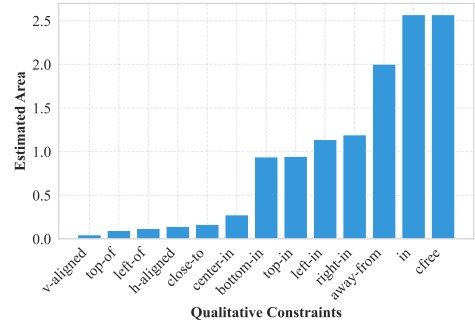

(a) **Failure rate of qualitative constraints**. The failure rate of each constraint is annotated on the bars. The red bars denote the number of failures, while the green bars denote success. Constraints are sorted by failure rate.

(b) **Estimated area occupied by qualitative constraints**. The order is highly similar to the order in (a), supporting the hypothesis that the smaller the defining region of a constraint, the harder it is to satisfy.

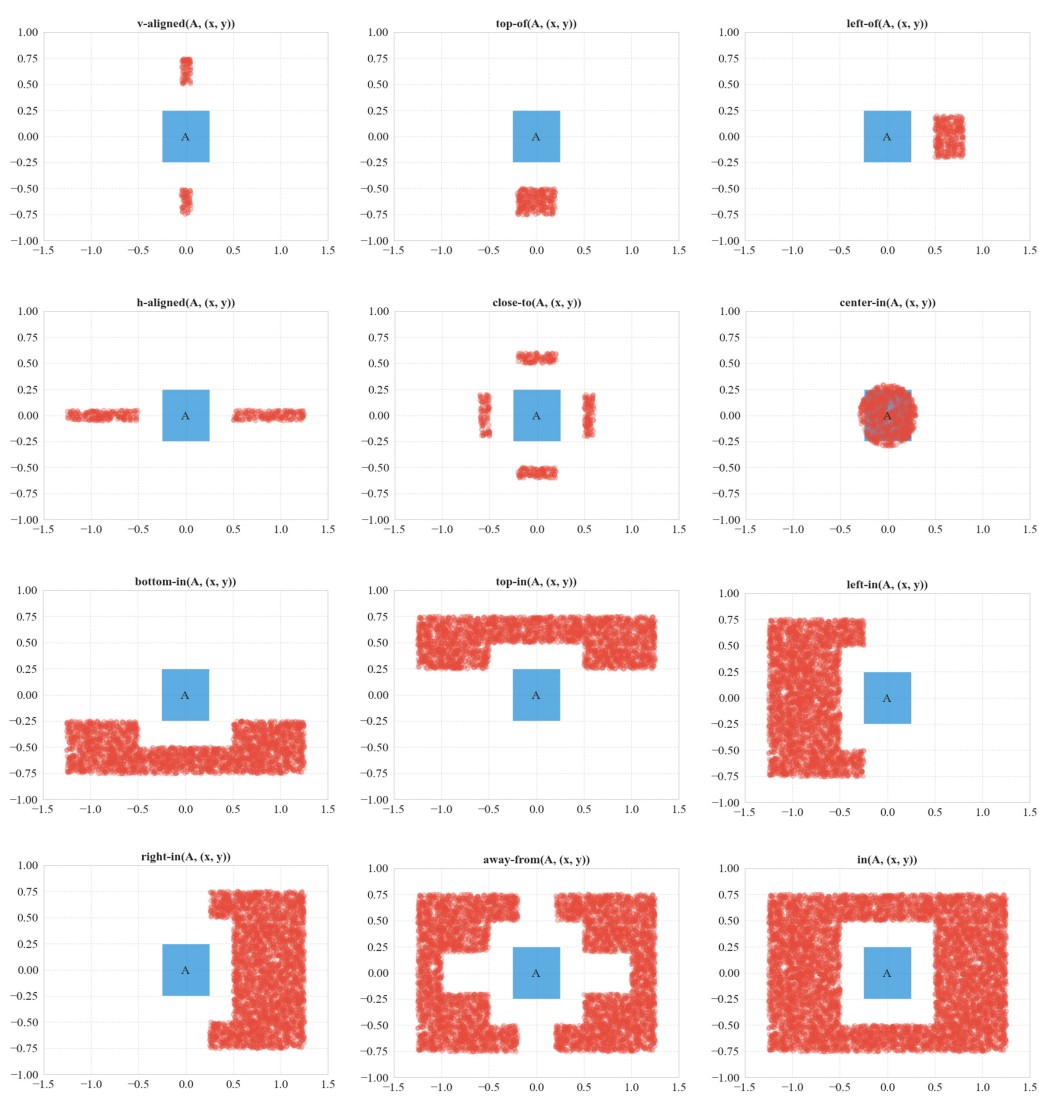

(c) Training data distribution for each constraint, in the order of increasing area of the defining region. Each point is the centroid for an object A that appeared in one *constraint*(A, B) inside the training dataset.

Figure 12: Analysis of how failure rate of qualitative constraints correlates with area of its defining region.

Figure 13: **Failure and success samples for Task 1: 2D Triangle Packing**. All solutions are generated by Diffusion-CCSP (ULA), trained on 30k data up to 4 objects.

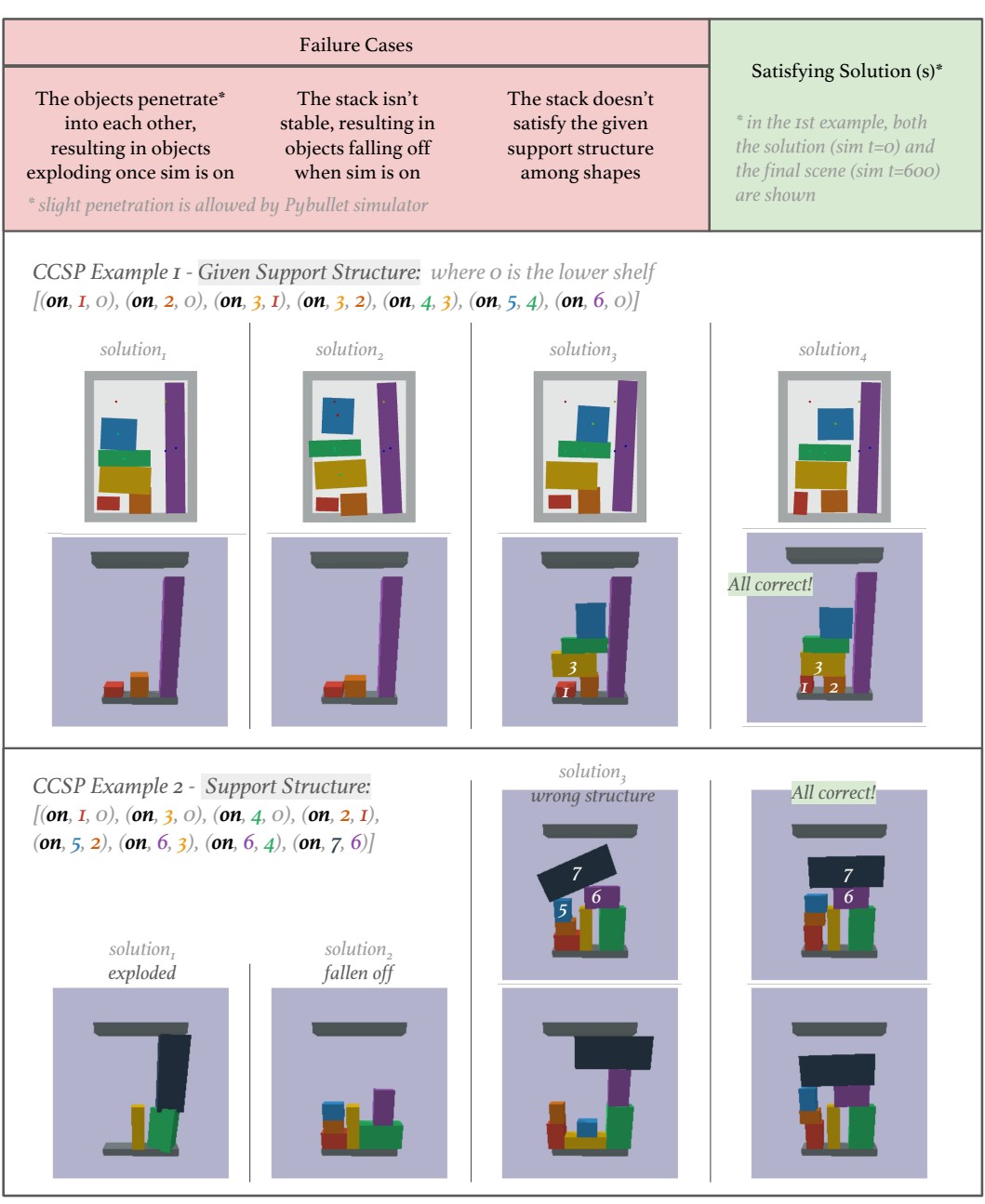

Figure 14: **Failure and success samples for Task 3: 3D Object Stacking with Stability Constraints**. All solutions are generated by Diffusion-CCSP (ULA), trained on 24k data up to 7 objects.

# D  Additional Sampler Comparison

In the main experiment section, we let Diffusion-CCSP and baselines generate 10 samples for each CCSP and check if all constraints are satisfied. Here we let Diffusion-CCSP generate 100 samples on 100 problems, and plot the number of problems it took Diffusion-CCSP to solve for each task. In a batch of 100 CCSPs, it takes on average 0.01-0.04 sec for Diffusion-CCSP (Reverse) to solve each CCSP while 0.06-0.19 sec for Diffusion-CCSP (ULA) to solve each CCSP, as shown in Table 2.

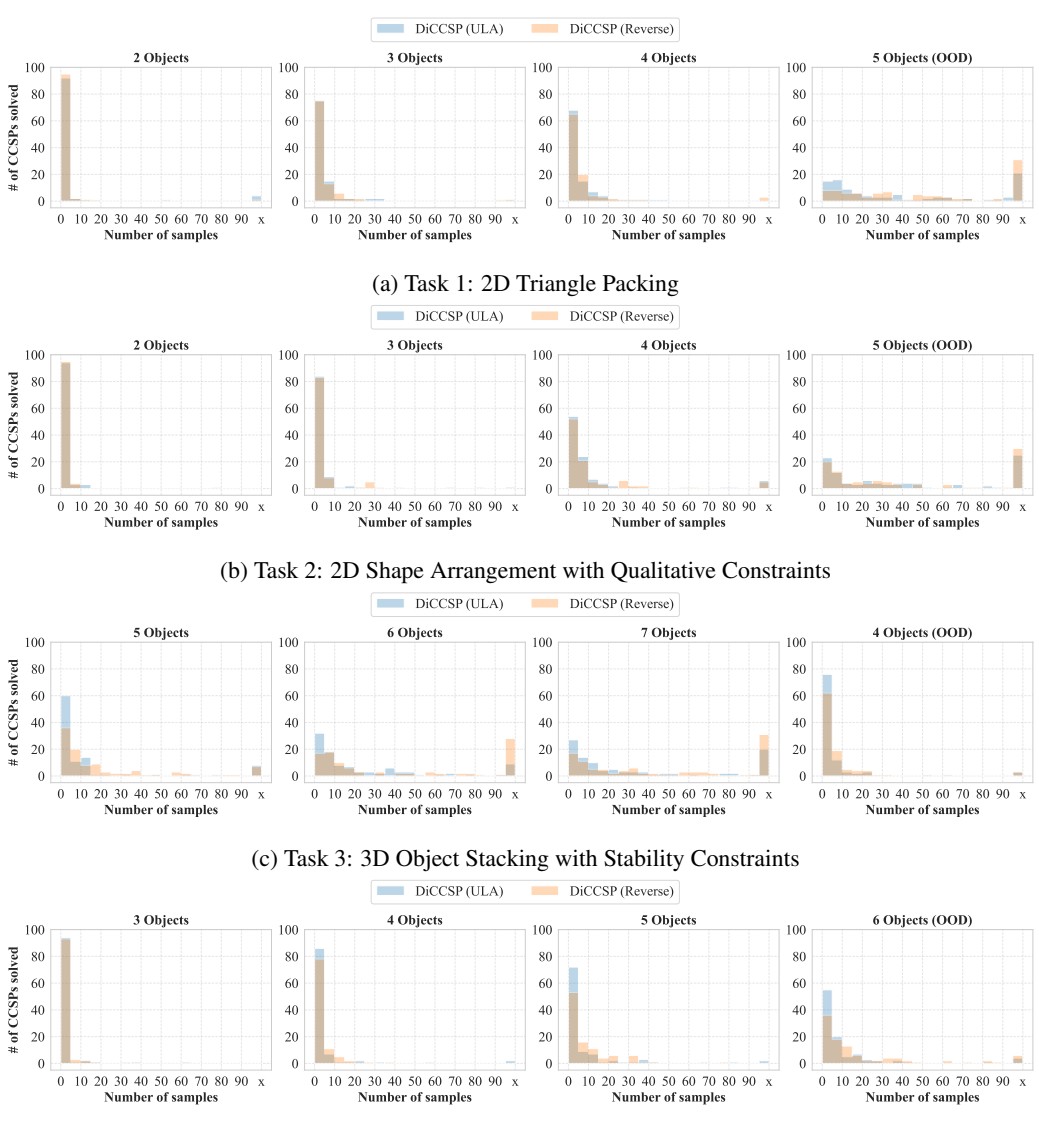

Figure 15: Number of Diffusion-CCSP runs it takes to solve 100 CCSPs. OOD means problems that are out of training distribution (i.e. include more objects than trained on). On the $x$-axis, tick x means problems not solved within 100 samples.

# E  Integration with Task and Motion Planning Algorithms

So far, we have presented a generic solution to solving constraint satisfaction problems that involve geometric and physical constraints. However, assuming that the constraint graph is given as input to

|                          | 2 Objects | 3 Objects | 4 Objects | 5 Objects |
|--------------------------|-----------|-----------|-----------|-----------|
| Diffusion-CCSP (ULA)     | 0.10      | 0.10      | 0.11      | 0.12      |
| Diffusion-CCSP (Reverse) | 0.01      | 0.01      | 0.01      | 0.01      |
| StructDiffusion          | 0.08      | 0.08      | 0.08      | 0.08      |

(a) Task 1. 2D Triangle Packing

|                          | 2 Objects | 3 Objects | 4 Objects | 5 Objects |
|--------------------------|-----------|-----------|-----------|-----------|
| Diffusion-CCSP (ULA)     | 0.17      | 0.18      | 0.19      | 0.19      |
| Diffusion-CCSP (Reverse) | 0.04      | 0.04      | 0.04      | 0.04      |

(b) Task 2. 2D Shape Arrangement with Qualitative Constraints

|                          | 4 Objects | 5 Objects | 6 Objects | 7 Objects |
|--------------------------|-----------|-----------|-----------|-----------|
| Diffusion-CCSP (ULA)     | 0.15      | 0.16      | 0.17      | 0.19      |
| Diffusion-CCSP (Reverse) | 0.01      | 0.01      | 0.01      | 0.02      |

(c) Task 3: 3D Object Stacking with Stability Constraints

|                          | 3 Objects | 4 Objects | 5 Objects | 6 Objects |
|--------------------------|-----------|-----------|-----------|-----------|
| Diffusion-CCSP (ULA)     | 0.06      | 0.06      | 0.07      | 0.07      |
| Diffusion-CCSP (Reverse) | 0.01      | 0.01      | 0.01      | 0.01      |
| StructDiffusion          | 0.09      | 0.09      | 0.08      | 0.08      |

(d) Task 4: 3D Object Packing with Robots

Table 2: Average run time (sec) of models in a batch of 100 CCSPs.

the algorithm. This approach can be directly used to solve particular tasks such as the pose prediction task in object rearrangements. Next, we illustrate how the proposed method can be integrated with a search algorithm to solve general task and motion planning (TAMP) problems, where the constraint graphs are automatically constructed based on the sequence of actions that has been applied and the goal specification of the task. For brevity, we will present a simplified formulation of TAMP problems. For more details, please refer to the recent survey [36].

Formally, given a space $\mathcal{S}$ of world states, a problem is a tuple $\langle \mathcal{S}, s_0, \mathcal{G}, \mathcal{A}, \mathcal{T} \rangle$, where $s_0 \in \mathcal{S}$ is the initial state (e.g., the geometry and poses of all objects), $\mathcal{G} \subseteq \mathcal{S}$ is a goal specification (e.g., as a logical expression that can be evaluated based on a state: $in(A, Box)$ $and$ $in(B, Box)$), $\mathcal{A}$ is a set of continuously parameterized actions that the agent can execute (e.g., pick-and-place), and $\mathcal{T}$ is a partial environmental transition model $\mathcal{T} : \mathcal{S} \times \mathcal{A} \rightarrow \mathcal{S}$. Each action $a$ is parameterized by two functions: precondition $pre_a$ and effect $eff_a$. The semantics of this parameterization is that: $\forall s \in \mathcal{S}. \forall a \in \mathcal{A}. pre_a(s) \Rightarrow (\mathcal{T}(s, a) = eff_a(s))$.

The goal of task and motion planning is to output a sequence of actions $\bar{a}$ so that the terminal state $s_T$ induced by applying $a_i$ sequentially following $\mathcal{T}$ satisfies $s_T \in \mathcal{G}$. The state space and action space are usually represented as STRIPS-like representations: the state is a collection of state variables and each action is parameterized by a set of arguments. Figure 16 shows a simplified definition of the pick-and-place action in a table-top manipulation domain.

A characteristic feature of TAMP problems is that the decision variables (i.e., the arguments of actions) include both discrete variables (e.g., object names) and continuous variables (e.g., poses and trajectories). Therefore, one commonly used solution is to do a bi-level search [37, 4]: the algorithm first finds a plan that involves only discrete objects (called a *plan skeleton*) and uses a subsequent CSP solver to find assignments of continuous variables, backtracking to try a different high-level plan if the CCSP is found to be infeasible. For example, consider the discrete task plan (also called *plan skeleton*): *pick-and-place*(A), *pick-and-place*(B). There are six parameters to decide: the grasps on A and B, the place locations of A and B, and the robot trajectory while transporting A and B. Based on the preconditions of actions and the goal condition of the task, we can obtain the constraint

```
;; pick up x and and place x to p.
action pick-place(x, g, p, t)
  pre: valid-grasp(x, pose[x], g)        ;; g is the grasp pose on x
       valid-traj(x, pose[x], g, p, t)   ;; p is the target pose of x
       forall z. cfree(x, z, pose[x], t) ;; t is the robot trajectory
  eff: pose[x] := p                      ;; object x will be moved

goal: in(A, C, pose[A], pose[C]) and in(B, C, pose[B], pose[C])
```

The precondition-effect definition of a continuously parameterized
action pick-place: pick and place object x.

Figure 16: **Illustration of a simple task and motion planning problem.** The domain contains only one action: pick-and-place of objects. It contains three preconditions: $g$ should be a valid grasp pose on object $x$, $t$ should be a valid robot trajectory that moves $x$ from its current pose to the target pose $p$, and during the movement, the robot and the object $x$ should not collide with any other objects $y$. If successful, the object will be moved to the new location. The goal of the task is to pack both objects into the target box without any collisions.

```
pick-place(A, gA, pA, tA)
 valid-grasp(A, pA0, gA)
 valid-traj(A, pA0, gA, pA, tA)
 cfree(A, B, pB0, tA)
 cfree(A, C, pC0, tA)

pick-place(B, gB, pB, tB)
 valid-grasp(B, pB0, gB)
 valid-traj(B, pB0, gB, pB, tB)
 cfree(B, A, pA, tB)
 cfree(B, C, pC0, tB)

Goal
 in(A, C, pA, pC0)
 in(B, C, pB, pC0)
```

(a) A partially specified plan skeleton and its corresponding constraints.

(b) The corresponding TAMP constraint graph derived from the plan skeleton. cfree constraints and geometric features are omitted for brevity.

Figure 17: Example of a partially specified plan skeleton in task and motion planning, together with the corresponding set of constraints. $p_{A0}$, $p_{B0}$, and $p_{C0}$ corresponds to the initial pose of objects.

graph. Therefore, each solution to the CCSP constructed based on the plan skeleton corresponds to a concrete plan of the original planning problem. By combining the high-level search of plan skeletons and our constraint solver Diffusion-CCSP, the integrated algorithm is capable of solving a wide range of task and motion planning problems that involves geometric, physical, and qualitative constraints.

# F   Learned Gradient Fields

We visualized the learned gradient functions for all qualitative constraints trained for Task 2 (Figure 18 column 3). The gradient functions for the first 6 constraints (from `in` to `bottom-in`) corresponds to the direction of changing the pose of a square shape of size 0.15 by 0.15 inside a container of size 1 by 1 given different initial poses. The gradient functions for the later 7 constraints (from `cfree` to `v-aligned` in Figure 19) corresponds to the direction of updating the pose of a square shape of size 0.15 by 0.15 inside a container of size 1 by 1 where another square shape of the same size has been fixed at pose $(0, 0)$. We show a pose sampled by Diffusion-CCSP for the target object in column 4. In column 5, we visualized the distribution of poses that satisfy each qualitative constraint in the training data. Note that the poses are normalized against the size of the container and the shapes of those objects are no longer 0.15 by 0.15.

We also computed the energy function as the difference between input noise and reconstructed output at the final step of diffusion (Figure 18 column 2). Note that the energy field is steep. Therefore the areas in the configuration space that meet the constraints exhibit considerably lower energy than areas that don't. As such, when constraints are combined, the solver tends to prioritize solutions that satisfy all constraints.

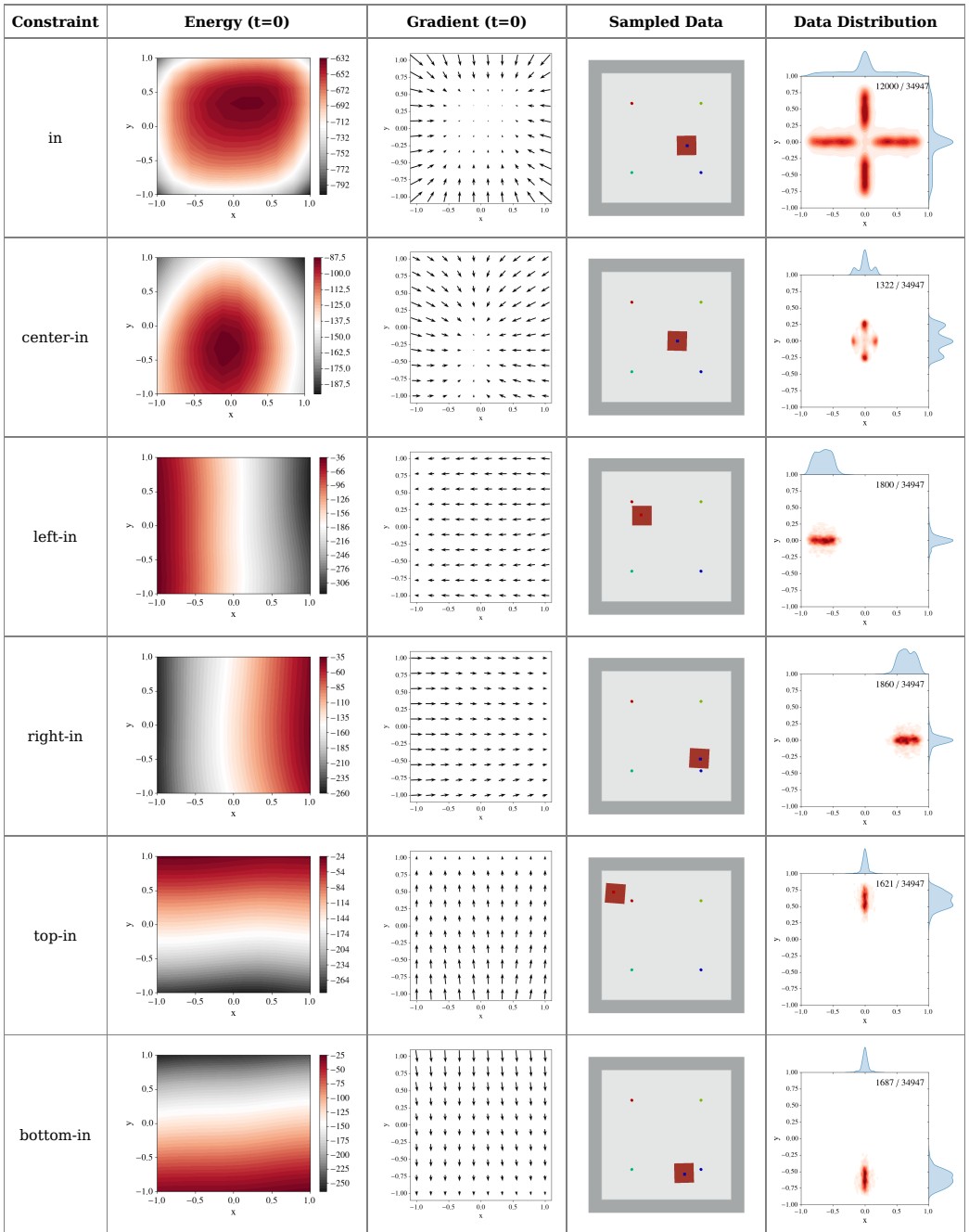

Figure 18: Visualization of energy and gradient functions of trained diffusion models for qualitative constraints (Part 1. Constraints between a shape and the container). **Column 2**: Energy fields inside a container of size 1 by 1. **Column 3**: Gradient functions that update poses in different time steps. **Column 4**: Poses sampled according to the given qualitative constraint. **Column 5**: Poses in the training data set with the given constraint, for 30,000 CCSPs that involve 2 - 4 objects that have different sizes. The number of constraints that appeared in the dataset is shown at the top right corner.

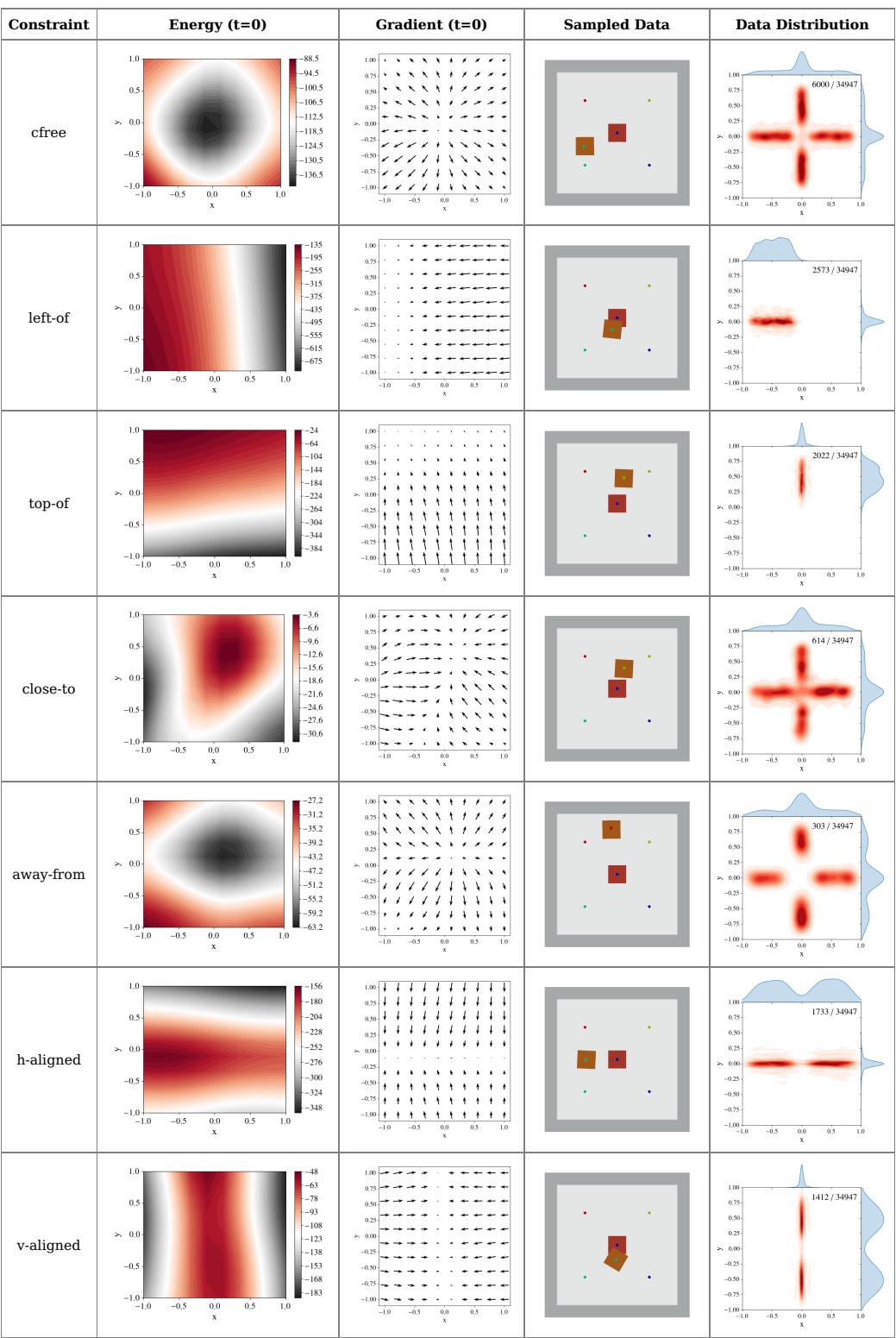

Figure 19: Visualization of energy and gradient functions of trained diffusion models for qualitative constraints (Part 2. Constraints between two shapes).

