# OpenReview forum: "Compositional Diffusion-Based Continuous Constraint Solvers"
_robot-learning.org/CoRL/2023/Conference — CoRL 2023 Poster_

### Official Review · Reviewer_wq1w · 2023-07-17

**Confidence:** 4
**Originality:** Good
**Technical Quality:** Very Good
**Clarity Of Presentation:** Very Good
**Impact:** 3

**Recommendation:**

Strong Accept: I recommend accepting the paper and will argue for my recommendation even if other reviewers hold a different opinion.

**Review:**

The paper reads well and is presented well in general. Although there has been some work in the context of composable diffusion models, and also representing constraints using diffusion models (decision diffuser), to the best of my knowledge I find the approach presented in this paper novel in terms of combining constraints with different types of modalities in robotics domain. My main concerns about this work: (1) still there is a gap from the proposed solution to having it functional in real world robotic applications. The sample complexity is concerning (10k - 30k); (2) the fact that the full model needs to be trained jointly presents some limitation in a sense that we also acquire skills not always in the context of a well defined task and solely based on curiosity. It is less clear to me how individual skill acquisition could be achieved more following a "divide and conquer" approach; (3) related to the previous point, it would have been nice to demonstrate how to fine tune when pre-trained skills and constraints are given a priori. Overall, I like the approach and find it very relevant to current trend in robotics.

**Quality Of The Limitations Section:**

Additional details required

**Questions For Rebuttal:**

1. Figures 1(c) and 1(d) do not really serve a purpose
2. ".. which involve both discrete and continuous parameters". Can you elaborate further on the "discrete" parameters?
3. Assumes fixed conditions (e.g., object features). What happens when the problem involves partial observability?
4. Sometimes collision is required for example when "tight packing" is required. How do we handle opposing constraints?
4. In example Figure 2(b) why pose A0 is required once given some other graspA?
5.  "It is important to note that these constraints are interdependent", I did not understand this point. The constraints could be in conflict.
6. It cries out for adding attention to the architecture.
7. Suggestion: "As object B is fixed" --> As pose of the object B remains unchanged
8. There is no Figure 3.c
9. What is "Object rearrangements"? (in introduction)
10. Could you precisely describe ULA?
11. Could you include any interesting failure cases? I am curious in what instances of the problem the model cannot generalize well.
12. Rearrange figure 5(c) for OOD (move it to the last column like every other experiment)
13. Could you address sample complexity issues when moving to real world robotic domains?
14. Please discuss some future work int he conclusion section if any.

**Robotics Focus:**

Highly relevant to robotics but no hardware experiments

**Summary Of Paper:**

This paper introduces a novel approach that leverages the compositional aspects of diffusion models to address robotics planning represented as continuous constraint satisfaction problems (CCSP). The fundamental idea revolves around representing a set of constraints as a computational graph, and employing modular diffusion models to handle various types of constraints. Leveraging the compositionality of these diffusion models, the approach can generate solutions by sampling from the joint model, ensuring satisfaction of the complete set of constraints. The authors showcase experimental results on simulated domains, demonstrating the model's generalization capabilities in handling novel combinations of constraints.

**Summary Of Recommendation:**

I fully understand the paper in terms of technical work and diffusion models in general, although I do not have a broad knowledge of the relevant work in space of robotic and control except for a few work. Having sad that, I would like to recommend for a weak accept to accept due to its relative novelty in robot planning, although I am concerned about sample complexity of this work when faced with real world robotics tasks.

---

### Official Review · Reviewer_88cB · 2023-07-19

**Confidence:** 5
**Originality:** Excellent
**Technical Quality:** Very Good
**Clarity Of Presentation:** Excellent
**Impact:** 4

**Recommendation:**

Strong Accept: I recommend accepting the paper and will argue for my recommendation even if other reviewers hold a different opinion.

**Review:**

The paper is very well written and easy to read. I personally was not familiar with the concept of CCSP and it was easy to understand thanks to the introduction. Framing the problem as the solution of the composition of multiple energy-based models is original and interesting and only few works [1,2] have explored this direction.

The problem they are tackling is an interesting one and it is properly introduced. The CCSP problem aims to solve a multi-constraint optimization problem. The solution should satisfy all the objectives at once. The authors have shown empirically the benefit of solving all the objectives jointly in contrast with sampling from a single constraint and trying to solve the rest of the constraints by rejection. This solution is in line with previous works [2] that have shown similar results for pick-and-place motion optimization problems.



**Strenghts**

1. The paper is very well written and easy to understand even for somebody that is not familiarized with CCSP, Composability in Energy Based Models or Diffusion Models.

2. The problem they are tackling is an interesting and important problem for robotics. Multiple robot tasks aim to find solutions finding trade-offs between multiple different objectives.


3. The paper presents an elegant and novel approach to CCSP, exploiting the composability of diffusion models.



**Weaknesses**


1. A known problem in EBM composition is the weighting of each energy in the optimization problem [2,3]. How did you tackle the problem of weighting the energy-based models?

2. Another known problem is the possible multiple local minimas in the optimization. A badly chosen initial seed might led to bad solutions. How do you tackle this problem?

3. The paper is not having any comparison with other methods that aim to solve similar problems. It would be interesting to observe if it possible with [1,4].



[1] Gkanatsios, Nikolaos, et al. "Energy-based models as zero-shot planners for compositional scene rearrangement." arXiv preprint arXiv:2304.14391 (2023).

[2] Urain, Julen, et al. "Se (3)-diffusionfields: Learning cost functions for joint grasp and motion optimization through diffusion." arXiv preprint arXiv:2209.03855 (2022).

[3] Liu, Nan, et al. "Compositional visual generation with composable diffusion models." European Conference on Computer Vision. Cham: Springer Nature Switzerland, 2022.

[4] Liu, Weiyu, et al. "Structdiffusion: Object-centric diffusion for semantic rearrangement of novel objects." arXiv preprint arXiv:2211.04604 (2022).


**Quality Of The Limitations Section:**

Additional details required

**Questions For Rebuttal:**

The questions are presented in the Weaknesses section. Main questions are:

1. How do you deal with the weighting of different energy components?
2. How do you choose your initial seed to tackle local minimas?

**Robotics Focus:**

Highly relevant to robotics but no hardware experiments

**Summary Of Paper:**

The paper tackles the problem of generating solutions for continuous constraint satisfaction problems (CCSP) in robotics. In particular, they focus in problems in which multiple modular constraints need to be satisfied jointly. This could be object-to-object constraints (such as two objects cannot place in the same place, due to collisions) or robot-to-object (plan trajectories collision-free)

This problem has been previously solved by learning generators for a single constraint and rejecting samples that do not solve previous constraints.

This paper proposes solving a joint-optimization problem. First, they represent the problem as a factor graph of constraints. Then, each constraint is represented with an energy-based model and learned with diffusion based techniques.

The authors claim generalization to novel combinations of known constraints and the application of the method for motion planning with both continuous and discrete actions.



**Summary Of Recommendation:**

I recommend an acceptance of the work. The paper tackles a very interesting problem for robotics and proposes and interesting solution.
Solving a CCSP problem with composition of energies is novel and it seems a promising direction.

---

### Official Review · Reviewer_drxW · 2023-07-19

**Confidence:** 2
**Originality:** Very Good
**Technical Quality:** Good
**Clarity Of Presentation:** Good
**Impact:** 4

**Recommendation:**

Weak Accept: I recommend accepting the paper, but will not argue for my recommendation if the majority of other reviewers have a different opinion.

**Review:**

The paper uses the compositional diffusion model to solve constraint satisfaction problems, which is novel and well-motivated. On four different domains, the proposed method shows strong performance compared to baselines. It generalizes well to novel combinations of known constraints. Some questions: could you run more seeds in Figure 5? or are the results consistent among different runs? Also, do you have more baselines to compare with in Fig 5 since you only compare your method with sequential sampling?

**Quality Of The Limitations Section:**

Limitations are addressed clearly

**Questions For Rebuttal:**

No.

**Robotics Focus:**

Highly relevant to robotics but no hardware experiments

**Summary Of Paper:**

The paper proposes a compositional diffusion continuous constraint solver, which learns to solve continuous constraint satisfaction problems. It uses constraint graphs as a unified view for different types of tasks and applies diffusion-model-based constraint solvers to find solutions. The proposed method is evaluated on four challenging domains and shows promising results.

**Summary Of Recommendation:**

The paper uses the compositional diffusion model to solve constraint satisfaction problems, which is novel and well-motivated. On four different domains, the proposed method shows strong performance compared to baselines. It generalizes well to novel combinations of known constraints. However, I am not very familiar with the topic and it is difficult for me to evaluate the significance of this work, so I give Weak Accept with low confidence considering the overall quality of the paper. I may change my score during the rebuttal.

---

### Comment · Area_Chair_5owi · 2023-08-11
**Discussion until Aug 15th**

I would like to thank the authors and reviewers and would like to encourage you to make best use of the discussion period which will end *Aug 15 at 11:59 PM PT*.

In particular, to the reviewers: you are highly encouraged to engage in additional discussions with the authors if there are any remaining questions you would like the authors to elaborate on further.

---

> ### Author Response · Authors · 2023-08-12
> **Thank you for your kind reminder**
>
> We've given our first round of responses to the reviewers' questions and suggestions. We're in the process of running more experiments based on their suggestion and will update through comments before the end of the rebuttal period.

---

### Decision · Program_Chairs · 2023-08-30

**Decision:**

Accept (Poster)

**Comment:**

I would like to thank the reviewers and authors for the fruitful discussion during the review process. The work proposes a novel approach to solve continuous constraint satisfaction problems in robotics which are of key relevance to the community. The approach of utilizing a compositional diffusion based approach appears to be novel and an improvement over the state of the art. The reviewers in particular highlight the following strengths of the work:

- the work is well written and approachable also for audiences new to CCSP
- the approach taken was considered elegant and novel
- the problem of CCSP in robotics was appreciated as being of high relevance
- the approach appears to generalize well to novel combinations of known constraints

Among areas of potential further investigation, the reviewers in particular listed
- potential remaining gap to real world deployment on robots
- sample complexity concerns
- questions around local vs global optimality during optimization
- investigation of fine-tuning of a pre-trained constraint solver

Overall the review consensus appears to indicate that this work may be a candidate for publication at CORL given in particular novelty and relevance to the community.